# Conditioning Machine Learning Models to Adjust Lowbush Blueberry Crop Management to the Local Agroecosystem

**DOI:** 10.3390/plants9101401

**Published:** 2020-10-21

**Authors:** Serge-Étienne Parent, Jean Lafond, Maxime C. Paré, Léon Etienne Parent, Noura Ziadi

**Affiliations:** 1Department of Soils and Agri-Food Engineering, Université Laval, Québec QC G1V 0A6, Canada; 2Agriculture and Agri-Food Canada, 1468 Rue Saint-Cyrille, Normandin QC G8M 4K3, Canada; jean.lafond@canada.ca; 3Département des Sciences Fondamentales, Université du Québec à Chicoutimi, 555, Boulevard de l’Université, Chicoutimi QC G7H 2B1, Canada; maxime_pare@uqac.ca; 4Departamento de Solos, Universidade Federal de Santa Maria, Camobi Santa Maria, Rio Grande do Sul 97105-900, Brazil; Leon-Etienne.Parent@fsaa.ulaval.ca; 5Agriculture and Agri-Food Canada, Quebec Research and Development Centre, 2560 Hochelaga Blvd., Québec QC G1V 2J3, Canada; noura.ziadi@canada.ca

**Keywords:** blueberry, crop modeling, plant nutrition, machine learning

## Abstract

Agroecosystem conditions limit the productivity of lowbush blueberry. Our objectives were to investigate the effects on berry yield of agroecosystem and crop management variables, then to develop a recommendation system to adjust nutrient and soil management of lowbush blueberry to given local meteorological conditions. We collected 1504 observations from N-P-K fertilizer trials conducted in Quebec, Canada. The data set, that comprised soil, tissue, and meteorological data, was processed by Bayesian mixed models, machine learning, compositional data analysis, and Markov chains. Our investigative statistical models showed that meteorological indices had the greatest impact on yield. High mean temperature at flower bud opening and after fruit maturation, and total precipitation at flowering stage showed positive effects. Low mean temperature and low total precipitation before bud opening, at flowering, and by fruit maturity, as well as number of freezing days (<−5 °C) before flower bud opening, showed negative effects. Soil and tissue tests, and N-P-K fertilization showed smaller effects. Gaussian processes predicted yields from historical weather data, soil test, fertilizer dosage, and tissue test with a root-mean-square-error of 1447 kg ha^−1^. An in-house Markov chain algorithm optimized yields modelled by Gaussian processes from tissue test, soil test, and fertilizer dosage as conditioned to specified historical meteorological features, potentially increasing yield by a median factor of 1.5. Machine learning, compositional data analysis, and Markov chains allowed customizing nutrient management of lowbush blueberry at local scale.

## 1. Introduction

Lowbush blueberry species (*Vaccinium angustifolium* Ait. and, to some extent, *V. myrtilloides* Michx.) are North American wild ericaceous species growing in upland acid sandy soils. The province of Québec, Canada, is among the world leaders in the production of lowbush blueberry [1]. Berry yields vary widely between 0.6 [2] and 8.9 Mg ha^−1^ [3], indicating high risk of production failure. Lowbush blueberry is managed over 2-year cycles where vegetative (or pruning) and fruit-bearing (or fruit-harvesting) years alternate. Flower bud initiation occurs during the vegetative year and impacts on crop productivity during the fruit-bearing year [4]. Fruit set depends on the number of flowers, pollination success, edaphic and managerial conditions, year, and clone [5], as well as nesting habitats of pollinators [6].

During the 2004–2009 period, low average yield of 1.9 Mg ha^−1^ impacted by adverse weather conditions affected the economic viability of most Quebec lowbush blueberry farms [7]. Snow cover, frost frequency, defrost and drought periods, flowering, weather variations, pollination, diseases, and maturation dates impact on lowbush blueberry productivity. Meteorological models have thus been developed to predict yields of lowbush blueberry and scout fields for pests [8]. Yield-impacting factors documented in large data sets can also be integrated into machine learning models [9,10,11,12].

Fertilizer trials on lowbush blueberry have been conducted as single nutrient N, P, and K experiments [3,13,14,15,16,17] as well as factorial N-P [18] and N-P-K combinations [2,19,20,21]. While ammonium-phosphate interaction may promote berry yields, lowbush blueberry appeared little responsive to added K [22]. Large variation in fertilizer dosage from 0.50 to 0.75 grower’s rate showed small impact on berry yield [23]. Little attention has been given to other elements [24]. Fertilization dosage and timing of application have not yet been optimized [17].

Nutrient management in lowbush blueberry production is conducted using soil and tissue tests. Where soil and tissue tests return opposite nutrient diagnoses, tissue tests are preferred [22]. Fertilization guidelines for lowbush blueberry are thus based mainly on tissue tests: soil tests are complementary. Tissue diagnosis as nutrient deficiency, sufficiency or toxicity is conducted by comparing each element to selected nutrient concentration ranges where crop productivity has been found to be adequate [25].

Regional nutrient standards derived using univariate descriptive statistical tests [26,27,28,29] may be hazardous because
nutrient variables are intrinsically multivariate—compositions should be interpreted as a whole, not as a collection of parts [30],regional standards disregard local conditions [22,31,32,33,34],descriptive statistical tests compare nutrient status of high and low yielders based on arbitrary yield threshold—they are designed to test differences, not to predict optimal compositions.

Our objectives were to predict yield of lowbush blueberry from a set of investigated feature-specific conditions and predict locally optimal fertilizer dosage, as well as optimal nutrient and soil compositions. We hypothesized that (1) soil chemistry, tissue nutrients, weather indices, and N-P-K fertilization affect berry yields, and (2) predictive models could optimize fertilizer dosage, as well as leaf nutrient and soil compositions under specified weather conditions.

## 2. Materials and Methods

### 2.1. Experimental Setup

Experimental sites were located in Normandin (48°50′ N, 72°32′ W), Saint-Eugène d’Argentenay (48°59′ N, 72°17′ W) and Labrecque (48°40′ N, 71°32′ W) in the Saguenay-Lac-Saint-Jean region, north-central Québec, Canada. The regional climate is at the edge between Dfb (warm summer continental or hemiboreal) and Dfc (subartic) [35]. The experimental areas were not irrigated as for most commercial lowbush blueberry fields in Quebec. Soils were sandy to sandy loam Spodosols developed on deltaic and eolian deposits [36]. There were 1504 observations collected from fertilizer trials conducted during the 2001–2011 period. The N, P, and K doses varied in the range of 0–90 kg N ha^−1^, 0–39 kg P ha^−1^, and 0–75 kg K ha^−1^.

While stands of lowbush blueberry were mixtures of phenotypically and genotypically variable clones [37] of *Vaccinium angustifolium* and *V. myrtilloides*, stands were dominated by *V. angustifolium*. The stands were fertilized every other year after pruning during the spring of the vegetative year to stimulate and support plant regrowth [17]. Because weeds strongly impact leaf nutrient concentrations and fruit yield of lowbush blueberry [38], all trials were realized in weed-controlled environments according to local recommendations [39]. Fields were managed for pests according to regional guidelines.

### 2.2. Soil and Tissue Analyses

Diagnostic tissues were collected at tip-dieback stage during the vegetative year [16,31,32,34,40]. Tissues were sampled in 50 m^2^ plot by combining the leaves of 25 randomly collected stems. Leaf samples were dried at 55 °C, ground to less than 1 mm using a Wiley mill, and digested in a solution of H_2_SO_4_ and H_2_O_2_ [41]. Digests were analyzed for total N, P, K, Ca, Mg, B, Cu, Zn, Mn, Fe, and Al. The N and P concentrations were quantified by automated colorimetry [Lachat Instruments (2005), QuickChem Method 13-107-06-2-E and QuickChem Method 15-501-3], and ICP-OES (Inductively coupled plasma-optical emission spectrometry) from Perkin Elmer (Waltham, Massachusetts) for other elements. Soil samples (0–20 cm), collected at the same time as tissue samples, were air-dried, 2-mm sieved, extracted using the Melhlich3 method [42], and analyzed for P, K, Ca, and Mg using ICP-OES. While soil nitrate and ammonium were extracted for half of the data set using KCl 2N then quantified by automated colorimetry, the present modelling was run across the whole data set excluding nitrate and ammonium. The pH was measured in water (1:1, *v*:*v*).

### 2.3. Meteorological Indices

Site weather data were downloaded from the closest (<50 km) Environment Canada meteorological stations using the weathercan R package version 0.3.4 [43]. Monthly weather indices computed from downloaded data are presented in Figure 1.

### 2.4. Investigative Models

We conducted exploratory analyses using two investigative models. The first investigative model considered seasonal mean temperature, total precipitations, and number of freezing days—days with minimum mean temperature <−5 °C [44]—between April and August inclusively in vegetative and fruit-bearing years. The second investigative model considered mean temperatures and total precipitations for phenological stages described by Fournier et al. [45] and presented in Table 1.

When conducting the predictive model, future weather is unknown. Thus, we fitted the predictive model to mean temperature and total precipitation data for phenological stages averaged over six years (or three crop cycles) preceding the season of observation. The phenology of *Vaccinium angustifolium* Ait. has been predicted from growing degree-days (GDD) using 0 °C [4] or 4.4 °C [5] as base air temperature from April 1st (day of the year 91). The GDD is commonly used in relation with pest management [7]. In this study, we tested mean temperature and growing degree days (>4.4 °C). After running preliminary models, we concluded that, compared to phenological stages and GDD, mean temperature offered more meaningful gradients across the whole season.

### 2.5. Statistical Analysis

#### 2.5.1. Isometric Log-Ratio

Raw concentration values were transformed into isometric log-ratios (*ilr*) to free compositional data from their total sum constraint (closure to measurement unit), and offer a sound framework to interpret tissue nutrient compositions [46]. Such framework is presented as a bifurcating tree or a mobile-and-fulcrum diagram based on nutrient interactions in living tissues [47] and soils [48]. Groups of variables were sequentially split until each group contain a single part (Figure 2). A filling value (Fv), computed by subtracting the sum of tissue elements from the total sum constrain (e.g., 100% or 1000 g kg^−1^), is included in balance diagrams to back-transform *ilr* balances to the more familiar concentration domain. Concentration values are shown at bottom and the balances at fulcrum of the bifurcating trees.

There are D-1 orthonormal balances in a D-part composition [49], each balance representing one degree of freedom [50]. Redundancy is accounted for by removing one degree of freedom producing resonance by altering proportions of components within a closed system. At one extreme, if two nutrients are fully synergistic or antagonistic, they carry the same information and one of them is thus redundant. However, no such nutrients exist as fully replaceable. The isometric log-ratios or log-contrasts between two subsets of components are computed as (1)
(1)$$ilr_{j}=\sqrt{\frac{r_{j}s_{j}}{r_{j}+s_{j}}}ln\left(\frac{g\left(c_{j}^{+}\right)}{g\left(c_{j}^{-}\right)}\right),$$
where, for the *j*th balance in [1…D − 1], D is the number of components, *r_j_* and *s_j_* are the number of parts on the left-hand- and right-hand side of the log contrast, respectively, *c_j_^-^* and *c_j_^+^* are the compositional vectors at the left-hand- and right-hand-side, respectively, and *g*() is the geometric mean function.

Computations were performed in the R statistical computing environment version 4.0.2 [51]. Leaf and soil nutrient concentrations were transformed into orthonormal nutrient balances or isometric log ratios [30] using the compositions package version 2.0-0 [52]. The Aitchison distance between a given nutrient composition and its target is a metric of interest to measure nutrient imbalance [46]. The Aitchison distance is a distance in the compositional space computed as the Euclidean distance between two equal-length compositional vectors transformed into *ilr* variables. The Aitchison distance depends on the number of components in the compositional vector and should be interpreted as a misbalance index compared to other distances computed from compositions with equal number of parts. Additionally, the ratio between each nutrient of an observation X and its target x indicates the direction of the misbalance.

#### 2.5.2. Analysis and Modelling

Investigative and predictive models relate yield to uncontrollable and controllable yield-impacting features. Yield variation could be explained by large differences in fertilization regimes, meteorological indices, as well as soil and tissue tests. While interactions between variables were likely to occur in excessively large numbers, they were not addressed in the present study to avoid over-fitting. The model thus addressed combinations of variables that are unique to each observation.

In a preliminary analysis, monthly weather effects on lowbush blueberry yields over a 2-years cycle led to model overfitting due to a too large number of features. We thus investigated the effects of leaf nutrients, soil nutrients, soil pH, NPK dosage and weather indices in two separate models (1) seasonal weather indices for 2-year cycle, and (2) monthly weather indices during the reproduction year only. We fitted Bayesian linear Gaussian regressions with vague priors using the rstanarm R package version 2.21.1 [53]. No model hierarchy (or random effects) was included to avoid over-fitting. All explanatory variables were centered at 0 mean and scaled to unit variance, allowing comparing slope coefficients on a common scale.

For the predictive model, the data set was split into 70% training and 30% testing subsets. All variables (outcomes and predictors) were centered to zero mean and scaled to unit variance based on the training set. We used a Gaussian process to predict yield, because this approach accurately returned smooth responses and acceptable accuracy in previous research on cropping system [9]. We did not attempt using other machine learning algorithms among the myriads available. The Gaussian process model was fitted to data using the kernlab package version 0.9-29 [54] with the caret modelling interface version 6.0-86 for R [55] with optimized hyper-parameters. The model fitted to training data was used to predict yield from features, some selected as varying and some selected as fixed, a process known as conditioning.

On the other hand, Markov chain random walk can optimize features such as tissue concentrations, soil analyses, and N-P-K fertilization under site-specific conditions. Markov chain is widely used in biology to model movement [56]. We fixed historical weather conditions for sequentially extracted combinations of randomly generated features for leaf nutrients, soil chemistry, and N-P-K dosages returning the highest yield in the neighborhood of optimal vector delivered by the previous sequence. This process is a Markov-chain random walk:use the model to predict yield from initial conditions,generate *n* random samples within a fixed radius around the point,to avoid extrapolation, compute the Mahalanobis distance between each random sample and the center and covariance of the training data set, then filter out random samples where the Mahalanobis distance is higher than a critical distance,use the model to predict yields from the remaining samples,extract the sample returning the highest yield,if yield is increased compared to the previous value, retain the current vector for the next round and shorten the radius by a factor—else, keep the previous vector for the next round, then increase the radius by a factor.

To show how this algorithm scans the multivariate space in search for higher yields, we used the R volcano data set [57] to generate a simplistic 2D space were the highest topography, modelled by a Gaussian process on a random sample of the data set, is approached from a starting point (white circle), as shown in Figure 3.

The optimization of leaf nutrient status was performed for each observation in our database. We randomly selected a sample from our database and looked for optimal leaf nutrient status, soil chemistry and fertilizer dosage under given weather conditions. Codes and data are available at git.io/JvQOa.

## 3. Results

### 3.1. Variability of Tissue and Yield Data at Regional State

Berry yields from experimental plots ranged between 0.6 and 13.8 Mg ha^−1^ in our data set and were more dispersed than lowbush blueberry yields published in other studies conducted in Maine, Québec, the Canadian Atlantic provinces, and Estonia (Figure 4).

### 3.2. Investigative Models at Regional Scale

#### 3.2.1. Effects over 2-Years Cropping Cycles

The first Bayesian linear regression with a gaussian response investigated the effects of leaf nutrients, soil nutrients, soil pH, NPK dosage and seasonal weather indices over 2-years on yields of lowbush blueberry. Posterior distributions of effects are shown in Figure 5.

The N and P fertilization averaged small positive effects, while K fertilization averaged marginal negative effects on berry yield. Seasonal total precipitation during both fruit-bearing and vegetative years increased berry yield. Seasonal mean temperature showed positive effect during the fruit-bearing year, but negative effect during the vegetative year. The number of freezing days during the fruit-bearing year markedly decreased yield but showed a small and uncertain effect on yield during the vegetative year.

The most impacting leaf nutrient balances were (1) the [B | Mg, Ca, K, P, N] balance, where higher concentrations of boron compared to macronutrients slightly decreased yield and (2) the [Fv | B, Mg, Ca, K, P, N] balance, where nutrient accumulation in tissues increased berry yield. Soil [Fv | Mg, Ca, K, P] and [Mg, Ca, K | P] were the most yield-impacting soil nutrient balances. The positive slope on the soil [Fv | Mg, Ca, K, P] balance indicated that higher yields were associated with higher nutrient levels in the soil. The negative slope of the soil [Mg, Ca, K | P] balance indicated that lower yields were associated with higher P concentrations relatively to cations K, Ca, and Mg in the soil. Low yields were associated with high soil pH.

#### 3.2.2. Effects during the Fruit-Bearing Year

A second investigative model substituted seasonal weather indices by weather indices at phenological stages for the year of experimentation (Figure 6).

As was the case of the 2-year cropping model, the effect of N-P-K fertilization in the fruit-bearing model was small compared to weather variables. The effect of mean temperature depended on developmental stage. Higher mean temperatures increased yields during the after fruit maturation and the flower bud opening stages, but decreased yields throughout the before bud opening and the fruit maturation stages. The effect was uncertain during the flower open stage. Precipitation effect also varied with the developmental stage. Higher precipitation increased yields during the flower open stage, but decreased yields during the flower bud opening and the after fruit maturation stages, with small and variable effects during the flower bud opening and fruit maturation stages. The number of freezing days, recorded only for the earliest development stage, showed a negative but uncertain effect on yield.

The [Fv | B, Al, Mg, Ca, K, P, N] leaf nutrient balance showed the most important positive effect among leaf nutrient balances, indicating that higher nutrient concentrations increased yield. The effect of the [B | Mg, Ca, K, N, P] balance was also positive, indicating that yield decreased with higher leaf B concentration. The [Mg, Ca, K | N, P] balance showed positive effect, indicating that higher N and P compared to K, Ca, and Mg leaf concentrations increased yield. While Redfield balance [P | N] showed positive effect, the effects of [Mg, Ca | K], [Mg | Ca] and [Al | B, Mg, Ca, K, N, P] were small and uncertain.

The effect of soil nutrient balances was also smaller than the effect of meteorological features. The most positive balances were greater soil nutrient supply capacity expressed as the [Fv | Mg, Ca, K, P], and higher K level in the cationic balance expressed as [Mg, Ca | K]. The most negative soil balance was [Mg, Ca, K | P], indicating excessive P level in the soil or insufficient concentrations of K, Ca, and Mg cations. Low berry yields were associated with high soil pH.

### 3.3. Predictive Model at Local Scale

While freezing days appeared to be important in both investigative models, they were not informative in the predictive model. Indeed, data exploration in Data Availability shows that the number of freezing days was inconsistent from year to year, making the 6-year average unreliable for yield prediction. The number of freezing days in April and May were thus removed from the predictive model.

The Gaussian process regression model returned root-mean-square-errors (RMSEs) of 1047 kg ha^−1^ in training and 1447 kg ha^−1^ in testing (Figure 7). Lower yields were predicted accurately while higher yields showed systematic deviation from the straight line. The classification mode could be useful to secure profitability above yield cut-off of 5000 kg ha^−1^. The accuracy of our model used in classification mode reached 83% on the testing set. The detection of low yielders was 91% accurate (positive predictive rate) compared to 53% for high yielders accurate (negative predictive rate). The low negative predictive rate is attributable to the large number of false positive specimens.

### 3.4. Portrait of Optimal Leaf Nutrients at Regional Scale

Because nutrient concentration ranges were feature-specific, we fixed no a priori optima for soil and tissue nutrient levels and looked for feature-specific optima. The Markov-chain algorithm applied to all weather conditions in the data set provided an overall portrait of predicted optimal leaf nutrient concentrations, which differed from concentration ranges suggested in the Canadian literature [16,21,32,33] (Figure 8). Note that the K range reported by Bouchard and Gagnon [33] for the same region was much lower than the distribution modelled from our data set.

Distributions of Aitchison distance and expected yield improvement by optimizing leaf nutrient levels are shown in Figure 9. The median Aitchison distance between nutrient balances of diagnosed tissue composition and optimal nutrient status was 0.50. Yield difference (potential yield minus initial yield) obtained where leaf nutrient compositions were perturbed from their initial composition to their optimal status varied widely with median value of 1773 kg ha^−1^, 1.5 times the yield of the diagnosed specimen for the specified combination of features. Expected yields reported in the data set for the specified combination of features were locally realistic compared to the arbitrarily expected high yields suggested at a regional scale.

The path to manage features returning the highest yield given a fixed set of local features was initiated by randomly sampling a low yielder (yield < 3000 kg ha^−1^, sample no. 1269), fixing weather features, then sequentially altering leaf nutrient levels, soil nutrient levels, pH and N-P-K dosage using the Markov chain algorithm. At each iteration of the Markov chain, we back-transformed leaf and soil nutrient levels from *ilr* variables to raw concentration values. We followed an optimal multivariate path towards optimum yield considering fixed historical weather conditions (Figure 10).

The Aitchison distance between the observed composition and the targeted composition obtained at the end of the Markov chain was 0.68 for leaf nutrients and 0.87 for soil nutrients. We also measured the size of the perturbation of nutrient composition between the observed leaf and soil nutrient compositions and the reference composition provided by the Markov chain algorithm as ratios their respective concentrations. The observed/target concentration ratios in Figure 11 show that leaf K, Al, and Mg concentrations appeared in relative excess in the diagnosed specimen compared to the successful specimen selected by the Markov chain algorithm, while B, P, and N appeared in relative shortage. Soils nutrients K, P, and Mg appeared in relative shortage while soil Ca and soil pH were near optimum.

## 4. Discussion

### 4.1. Model Features

Agroecosystems viewed as Humboldtian agricultural production units [12] require assembling local features to make reliable predictions on system’s performance. Indeed, the concept of optimum nutrient management under heroic assumptions of otherwise optimum conditions [61] may fail at local scale where genetic and environmental conditions may vary widely [62]. We used leaf nutrients, soil nutrients, pH, and weather data as features to predict yields of lowbush blueberry for stand mixtures of *Vaccinium angustifolium* and *V. myrtilloides*, using a Gaussian process machine learning model. By conditioning the model on the selected uncontrollable features such as weather historical data, and allowing other features related to plant nutrition management to vary, we could elaborate crop management recommendations at local scale. Where the model was conditioned on weather features, the localized model predicted that optimized plant nutrition and soil chemistry at local scale could increase berry yields substantially (Figure 9B).

### 4.2. Weather Indices

For the 2-year cycles and fruit-bearing year models (results presented in Figure 5 and Figure 6), weather features dominated largely yield potential of lowbush blueberry in Quebec. Developmental stages were sensitive to precipitation. While total precipitation at flowering stage showed positive effect, heavy precipitation may decrease pollination activities and increase the incidence of plant fungal diseases [63]. Nevertheless, plant-pollinator networks are impacted by rainfall patterns [6]. Heavy precipitation affects pollinators’ success through nectar dilution, pollen degradation, volatile removal, etc. At the other extreme, where precipitation is too low, irrigation is required to avoid shifting from reproductive to vegetative growth [64]. While favorable weather conditions for pollination activities during the month following pollination (July) are critical to reach maximum yield of lowbush blueberry, yield predictions were inconsistent based on meteorological features alone [63]. Adding soil and tissue nutrient features and phenological stages, our predictive reached a RMSE of 1447 kg ha^−1^ in testing. The acceptability of this precision is a professional decision considering risks of committing errors. The most comparable metric found in the literature is a classification accuracy: once used as a classification model with a yield cutoff of 5000 Mg ha^−1^, our model reached an accuracy of 83%, a value comparable to model accuracy for other fruit crops [65].

### 4.3. Fertilization

Response of lowbush blueberry to N, P, and K fertilization was small. Nevertheless, lowbush blueberry may respond positively to added N and P [18,66,67,68]. The weak negative response to K fertilization is attributable in part to antagonism between cationic macronutrients [69]. As wild species, in contrast with domesticated species, lowbush blueberry responds slowly to nutrient supply [34] and could constrain its growth rate to available resources [70]. Moreover, nutrient accumulation in reserve tissues could be remobilized during the following years, as reported for fruit trees [71] and vines [72].

While regional N recommendation averaged 45 kg N ha^−1^ [38], nitrogen dosage appeared to be highly site-specific. A fertilizer trial in Nova Scotia, Canada, indicated optimum dosage of 35 kg N ha^−1^, 40 kg P ha^−1^, and 30 kg K ha^−1^ [68]. Predicted fertilizer dosage for the low-fertility soil in our case study (Figure 10A, 61 N ha^−1^, 14 kg P ha^−1^ and 32 kg K ha^−1^) departed from current ranges of 25–60 kg N ha^−1^, 7 to 9 kg P ha^−1^ and 16–20 kg K ha^−1^ [73]. The N requirement up to 61 kg N ha^−1^ could be split between the spring of the vegetative year and the spring of the fruit-bearing year [73]. It should be emphasized that N fertilization may decrease berry quality, as shown by linear decrease of total polyphenols upon N additions of 0, 30 and 60 kg ha^−1^ to highbush blueberry [74]. The response to added N may also be modulated by competition with weeds [38].

In contrast with N, the response to P fertilization was found to be small [75]. The fact that the soil [Mg, Ca, K | P] balance impacted negatively on berry yield indicated that feature-specific corrective measures should be adopted to re-establish the soil P balance (a notion different than P budget) and avoid excessive soil P accumulation. The P fixation by oxy-hydroxides of Fe and Al at low pH values reduces P fertilizer-use efficiency in the acidic P-fixing podzolic soils used for lowbush blueberry production [76]. However, making P fertilizer application based on soil P fixing capacity alone can result in wrong decisions [77]. Soil pH values exceeding 5.2 can decrease the yield of lowbush blueberry [73].

In our study, increasing the [B | Mg, Ca, K, P, N] tissue balance increased berry yield (Figure 5 and Figure 6). Since 2000 in Quebec, shoot tip abortion is prevented by applying 0.7 kg of B ha^−1^ at each crop cycle [73]. While boron application is recommended to avoid boron shortage in blueberry plants [78,79], boron over-fertilization may reach toxicity levels. Indeed, leaf B concentration may increase by 4–5 folds with B application over control [80]. Because B has the narrowest range between deficiency and toxicity, cells are highly permeable to boron and boron is highly soluble at low soil pH values, excessive boron supply may be detrimental [81]. Boron is managed to reach optimal growth conditions based on leaf analysis and proper nutrient balances to avoid excessive B applications.

Soil test B using hot water as extracting solution (B_water_) is not commonly conducted for routine soil analysis in Quebec. However, the relationship between Mehlich3-B and hot-water B was found to be close (R^2^ = 0.98) after adding soil pH in water and organic carbon in the equation [82]. Suggested soil fertility classification for B_water_ in Quebec was as follows: 0.00–0.23 mg B_water_ kg^−1^ for low fertility class, 0.23–0.58 mg B_water_ kg^−1^ for medium class, and 0.50–3.70 mg B_water_ kg^−1^ for high fertility class. In comparison, Brdar-Jokanović [81] reported (1) boron shortage in soils containing less than 0.5 mg B_water_ kg^−1^, optimal level of 0.7 mg B_water_ kg^−1^ and threshold toxicity level of 1.5 mg B_water_ kg^−1^ for sensitive plants, and (2) 0.0–0.2 mg B_water_ kg^−1^ for low fertility class, 0.21–0.6 mg B_water_ kg^−1^ for medium class, 0.61–1.1 mg B_water_ kg^−1^ as high soil test B, 1.2–3.0 mg B_water_ kg^−1^ as very high soil test B, and >3.0 mg B_water_ kg^−1^ as toxic soil test B. The Mehlich3 method is commonly used in Quebec routine laboratories [83]. The B_Mehlich3_ soil fertility classes corresponding to the B_water_ fertility classes were found to be 0.00–0.65, 0.65–1.03, and 1.03–12.70 mg B_Mehlich3_ kg^−1^, respectively [82]. A survey of 50 blueberry farms of the Lac-St-Jean region showed soil test B_Mehlich3_ values varying in the range of 0.2 to 0.9 mg B_Mehlich3_ kg^−1^, mostly located within the medium B_Mehlich3_ fertility class. Although soil test B_Mehlich3_ has not been conducted during the present study because only tissue tests have been used to diagnose nutrient problems in the lowbush blueberry production by that time. Tissue test integrates all factors that regulated plant nutrition [25]. Soil test B_Mehlich3_ classification [82] could be useful to avoid B over-fertilization of blueberry stands.

The Al concentration in plant tissues may be problematic in acid soils due to high Al toxicity [84]. Blueberry soils of the region were found to contain 893-1430 mg Al_Mehlich3_ kg^−1^ [75], within the range of 496–2424 mg Al_Mehlich3_ kg^−1^ reported for Quebec soils [83]. The leaf Al concentration may also depend on soil pH because (1) Al tends to be mobilized in soils at pH lower than 5.5 [84,85] and even more at pH values less than 5.0 [86], and (2) lower pH is associated with higher berry yields. Foliar tissues of lowbush blueberry normally contains 50–110 [32], up to 400 [87] mg Al kg^−1^ compared to 400–760 mg Al kg^−1^ in rhizomes [87], indicating genetic control on Al translocation from the belowground to the aboveground plant parts. In Al-tolerant species, Al may stimulate growth by preventing Cu, Mn and P toxicity and acting as fungicide against certain types of root rot [69]. In our study, the Markov chain random walk indicated optimum foliar Al concentration of 45 mg Al kg^−1^ (Figure 10B) in a locally diagnosed specimen where simulated optimal pH was 4.6—an Al level close to the median value of its distribution in our data set (51 mg Al kg^−1^, Figure 8).

### 4.4. Agronomic Features Optimisation

Open ecosystems have several site-specific unexplained or uncontrollable sources of variation. While the R^2^ of the Gaussian process regression was rather low (R^2^ = 0.46, root-mean-square-error of 1447 kg ha^−1^) in testing, accuracy reached 83% in classification mode, a value comparable to other fruit crops [10,47,88,89,90]. The regression mode can compare current yields to modelled yields under optimized nutrient management. The classification mode can customize yield targets at field scale. In this paper, we challenged currently used regional tissue nutrient concentration ranges for the following reasons.
Regional guidelines deny the importance of local conditions on plant epigenetics.A collection of reference ranges relies on the assumption that the space of successful nutrient dosage and leaf and soil compositions have the shape of hypercube. As illustrated by Parent [46], the shape of such space is irregular and blob- or cloud-like.Arbitrarily delimiting the contours of a prosperous or successful region [10,91,92] should be avoided.According to Parent [46], interpreting a perturbation between a nutritionally imbalanced specimen and its optimum or successful target “should be done with a multivariate and compositional data perspective in mind. This implies that (1) a univariate or an incomplete multivariate perspective (e.g., focusing on extreme excesses and deficiencies) could miss a high yield region (a parachutist adjusting her fall following only one axis will likely miss the enchanting island and fall into the sea) and (2) changes of concentrations in a closed system are relative, i.e., increasing the concentration of a component will inevitably decrease the concentration of at least another one”.

Instead of diagnosing ranges of leaf nutrients, soil test values and fertilizer dosage at high-yield level, as for the agronomic interpretation methods used so far, we followed a Markov chain random walk towards optimal values conditioned by local weather features to customize critical ranges at local scale. Those results emphasize the need to focus on crop management at local scale, regularly updating the data set with experimental and observational data.

## 5. Conclusions

Our investigative models related berry yields to soil and tissue tests, weather indices, and N-P-K fertilization. Relative P excess in the soil, too high soil pH, and relative B excess in the tissue mass impacted berry yield negatively. The Gaussian process model predicted yield from leaf nutrient composition, soil tests, fertilizer dosage, and weather conditions. We elaborated a novel in-house Markov chain algorithm to follow a path from current cropping environment to an environment that maximizes yield along improving leaf nutrient compositions, soil chemistry, and fertilizer dosage given historical weather indices. Such a modelling approach is the first ever to optimize soil and tissue compositions and fertilizer dosage simultaneously, while providing realistic yield expectations at local scale. Obviously, present nutrient management concepts supported by soil and tissue tests alone such as regional tissue nutrient standards, buildup and maintenance, cation saturation ratios, or nutrient sufficiency levels should be revisited to optimize not only crop yields, but also product quality and environmental impacts by making economically- and environmentally-wise fertilization decisions at local scale. Large data sets can be processed using machine learning and Markov-chain optimization to develop reliable solutions at local scale under varying scenarios of feature combinations, including climate change.

Unlike tissue concentration ranges and soil fertility classification based on descriptive statistics and dichotomous decisions, machine learning models can predict yield from specified combinations of features that are documented in large data sets. The lowbush blueberry data set could be augmented and updated regularly to tackle the source of yield variation and implement means to sustain production of lowbush blueberry by rebalancing nutrients at local rather than regional scale. Because growers collect large amounts of local data such as soil and tissue tests and berry yield and quality data, and as more soil and climatic data become accessible, the lowbush blueberry data sets can grow up rapidly. Models conditioned to local features for predicting yield, crop quality, and ecological impacts should be adopted where sufficient data are available.

## Figures and Tables

**Figure 1 plants-09-01401-f001:**
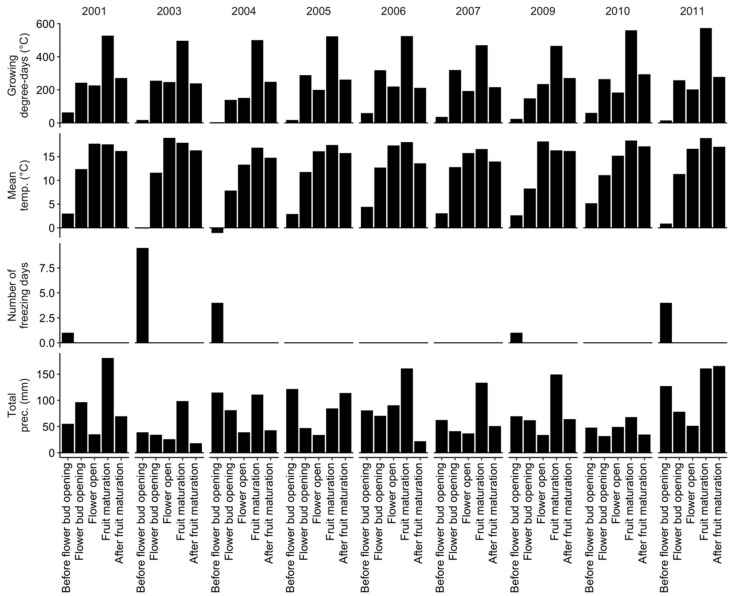
Mean weather indices computed across sites from 2001 to 2011, excepting 2002 and 2008 when no data have been collected.

**Figure 2 plants-09-01401-f002:**
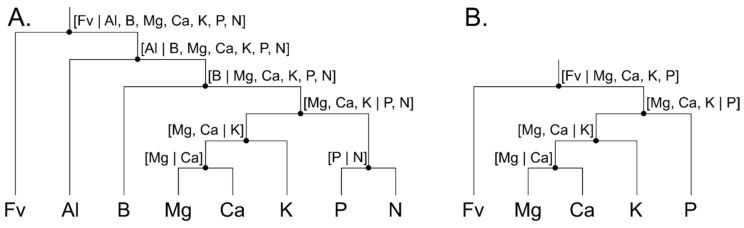
Balance diagram to transform (**A**) nutrients to nutrient balances and (**B**) soil nutrients to soil balances. Fv is the filling value.

**Figure 3 plants-09-01401-f003:**
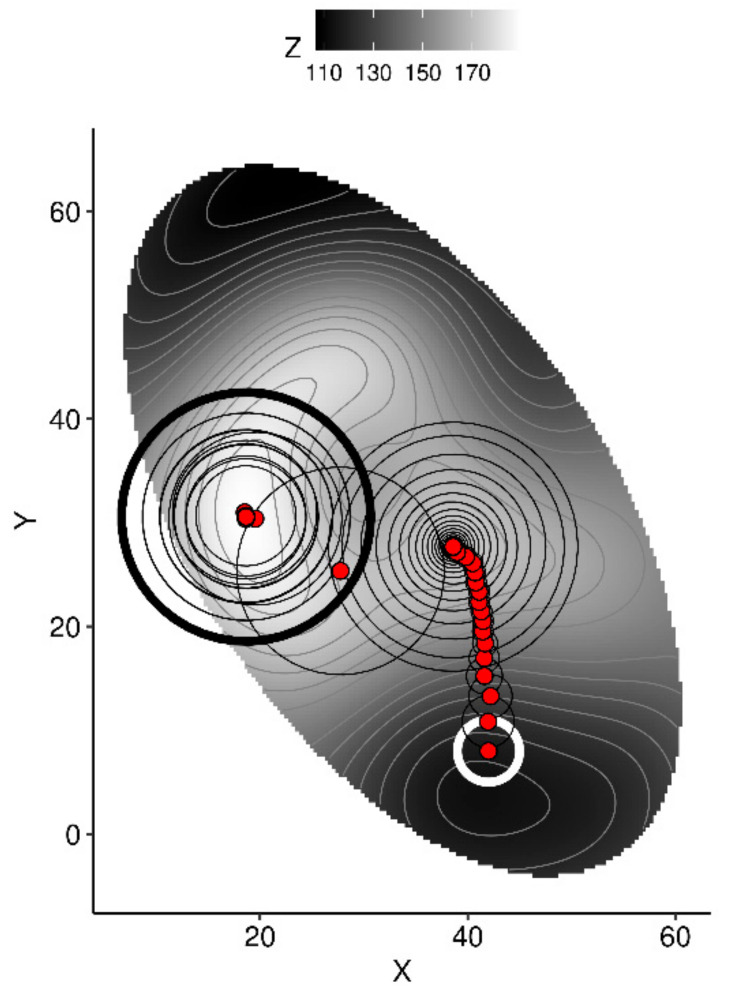
Two-dimensional representation of the algorithm scanning XY coordinates to draw the path to higher Z topography as a metaphor for scanning tissue nutrient balances that augment yields. The red dot started at (42, 8) with a radius of 3 (thick white circle), moved with a decreasing radius to reach a local optimum where radius was increased until finding another point from which it continued scanning until the maximum of iterations was reached (thick black circle).

**Figure 4 plants-09-01401-f004:**
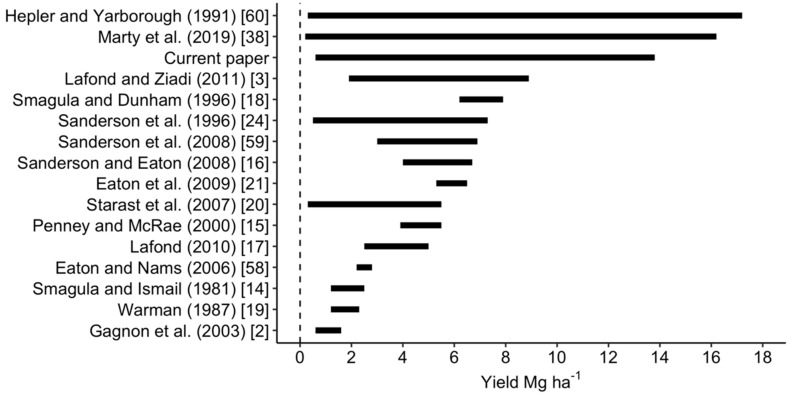
Yield ranges of lowbush blueberry reported in the literature compared to yield range in the present study [2,3,14,15,16,17,18,19,20,21,24,38,58,59,60].

**Figure 5 plants-09-01401-f005:**
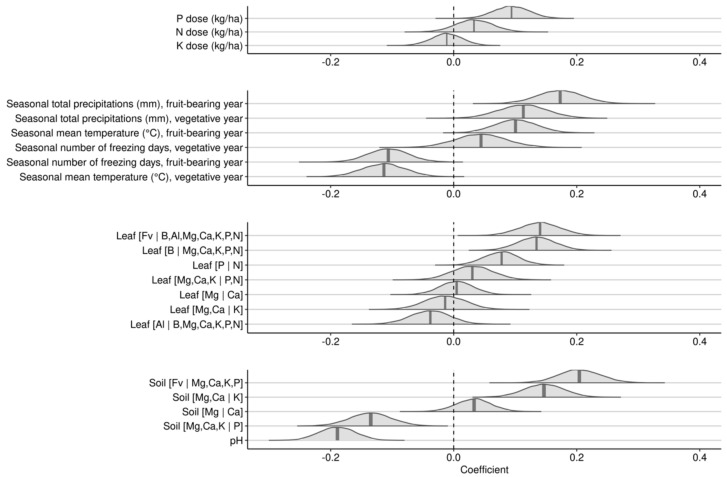
Posterior distributions of coefficients of scaled variables against berry yield for the 2-year cycle model.

**Figure 6 plants-09-01401-f006:**
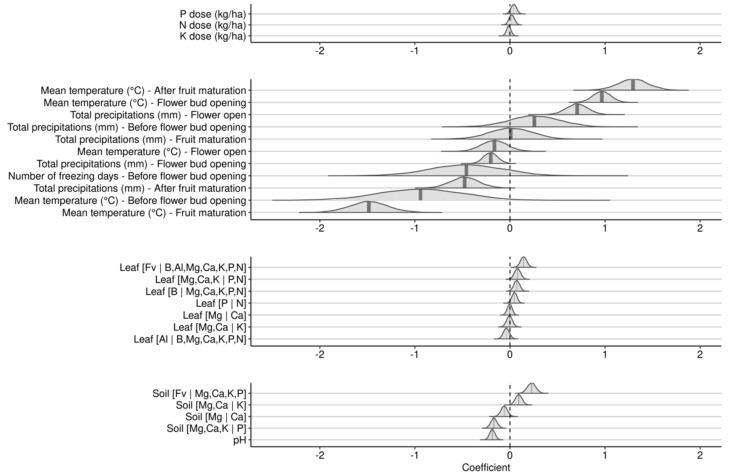
Posterior distribution of coefficients of scaled variables against berry yield for the fruit-bearing year model.

**Figure 7 plants-09-01401-f007:**
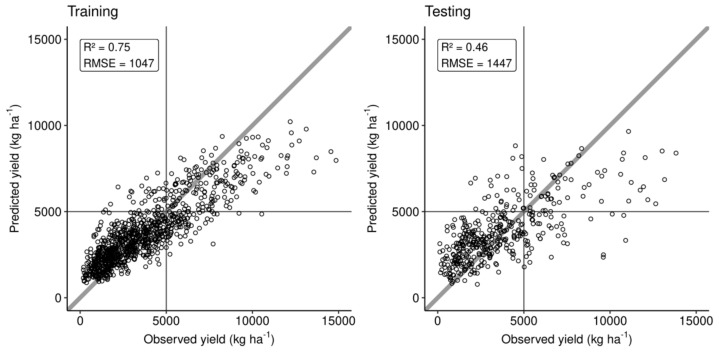
Performance of the predictive Gaussian process model shown as prediction against observed in training and testing data sets.

**Figure 8 plants-09-01401-f008:**
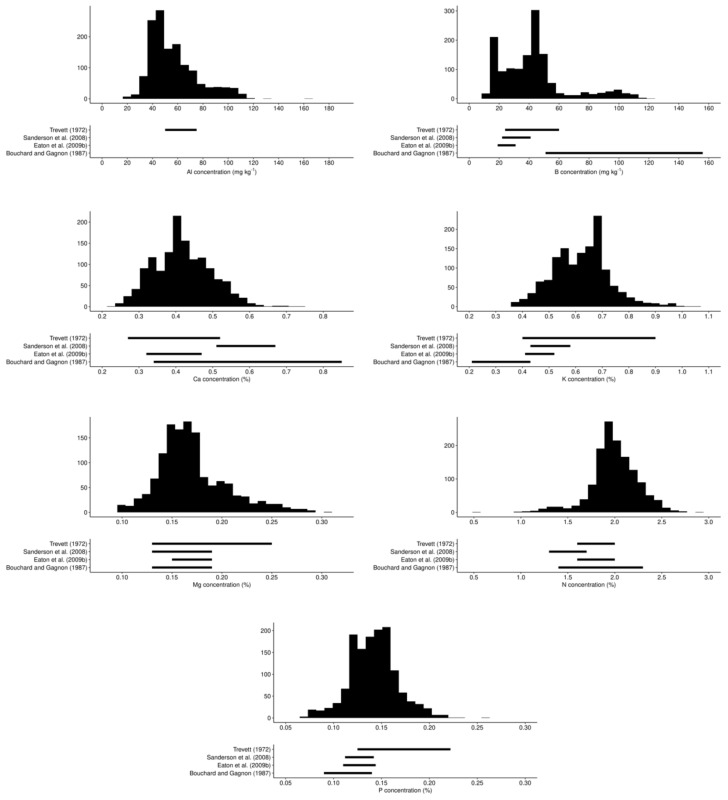
Distributions of optimal concentrations in leaf tissue of blueberry in the present study compared to ranges reported in the Canadian literature [16,21,32,33].

**Figure 9 plants-09-01401-f009:**
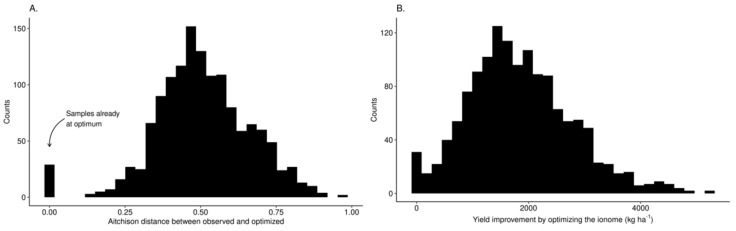
Distributions of (**A**) optimal Aitchison distances and (**B**) computed yield improvements by optimizing leaf nutrient status.

**Figure 10 plants-09-01401-f010:**
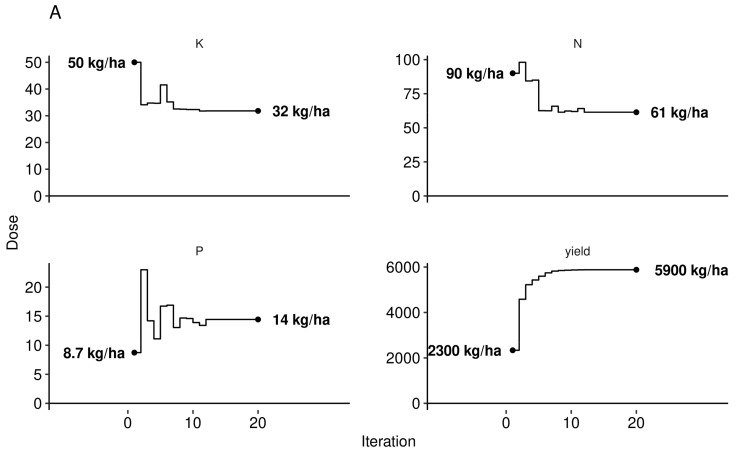
Markov chain searching for (**A**) N-P-K dosage (**B**) tissue concentration ranges and (**C**) soil chemistry matching the highest yield (shown in (**A**)) under given historical weather conditions of the randomly selected sample no. 1269. Constrained paths represent minimum and maximum values in the training data set and avoid modelling extrapolations.

**Figure 11 plants-09-01401-f011:**
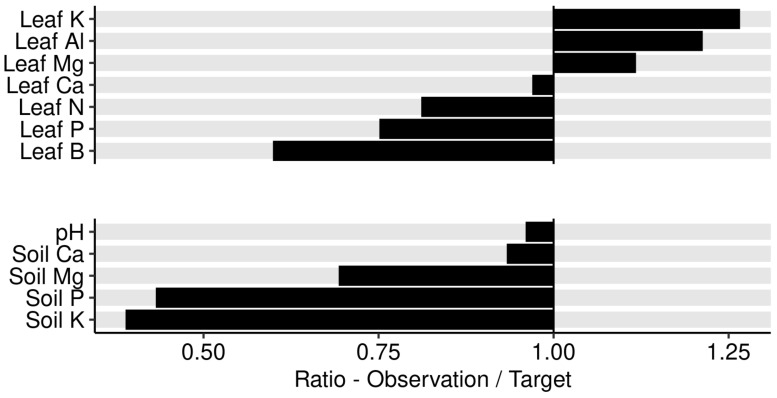
Ratio between leaf nutrients in sample no. 963 and the optimal composition found at the end of the Markov chain algorithm.

**Table 1 plants-09-01401-t001:** Phenological stages of lowbush blueberry in the Lac-St-Jean region, North-Central Quebec, Canada. Data from [45].

Phenological Stage	Julian Day	Calendar Dates
Before flower bud opening	[92 to 125]	1 April to 5 May
Flower bud opening	[126 to 163]	5 May to 11 June
Flower open (Pollination period)	[164 to 180]	12 June to 28 June
Fruit maturation	[181 to 220]	29 June to 7 August
After fruit maturation (Harvest)	[221 to 244]	7 August to 31 August

## Data Availability

Codes and data are available at git.io/JvQOa.

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
