# Peer review of "Conditioning Machine Learning Models to Adjust Lowbush Blueberry Crop Management to the Local Agroecosystem"

_plants, 2020, doi:10.3390/plants9101401_

Round 1

Reviewer 1 Report

Please see attachment with comments on the manuscript

Reviewer 2 Report

Comments for Plants-953362

Overall, the paper is written well. After going through the manuscript, the following issue were observed

From the template used, I am confused whether the authors are submitting to MDPI-Plants or MDPI sensors.

Abstract

In first sentence “yield-limiting features” should be changed to “yield-limiting variable”

Introduction

More emphasis should be given to the climate factors affecting crop yield by consulting the latest papers

Discussion

The discussion portion needs more attention and more critical analysis of the results is required by consulting the latest papers from 2020

Reviewer 3 Report

The modeling of your manuscript is novel, and have a value to publish in Plants. However, you need major revision before acceptance. Especially, the content of discussion was quite poor. Unfortunately, I could not understand your new findings from the chapter of 4.1 and 4.2. In the case of the chapter of 4.3, the main contents were quotation of literatures, and information of your results was insufficient. You need to emphasize the beneficial trends from your results, and discuss to estimate your results by the comparison of previous literatures. Moreover, your conclusion was inarticulate. In my opinion, there is possibility that simple regression analysis may leads clear conclusion. You should conclude the novel findings by use of Markov chain or random walk.

 Also, I would like to indicate several weak points as following:

  1. According to your results, leaf N and B are important to regulate the yield of blueberry. However, there was no data of N and B in soil. Also, there was no data of Al in soil. I think that concentrations of N, B and Al in soil is quite important to regulate the yield of blueberry. I recommend to add the data of N, B and Al in soil, and recalculate the model.

  1. In introduction, induction of Markov chain or random walk was not logical. Some readers cannot understand that the utilization of Markov chain or random walk is useful to estimate nutrient status. You must quote the literatures that estimate nutrient status by Markov chain or random walk.

  1. I could not understand the reason why you made two models written in 3.2.1 and 3.2.2. In my opinion, effects during the fruit bearing year was logical from the phenology of blueberry. In contrast, I could not the reason why you estimate over 2-years cropping cycles. The results of two models were almost same between Figure 5 and 6. Unless you explain the essential reason, you should omit Figure 5.

Round 2

Reviewer 1 Report

The paper has been significantly improved according to reviewer suggestions

Author Response

Thank you for the review.

Reviewer 2 Report

The authors have incorporated the suggested changes

The paper may be considered for publication 

Author Response

Thanks for the review.

Reviewer 3 Report

On the revised manuscript, I admitted your efforts for the improvement of discussion. On the introduction of Markov chain or random walk, it was better to explain in the introduction. However, there was no problem to explain Markov chain and random walk in the Materials and Methods. Therefore, I think that your manuscript is acceptable in progress.

 However, you did not explain the reason why you made two models (3.2.1 and 3.2.2) in the text. You must explain the reason why you show the two models in Materials and Methods.

Author Response

Reviewer 3 asked for a minor revision.

R3. You did not explain the reason why you made two models (3.2.1 and 3.2.2) in the text. You must explain the reason why you show the two models in Materials and Methods.

In the new version, we explained why we computed two models.

Manuscript, lines 170-174. In a preliminary analysis, monthly weather effects on lowbush blueberry yields over a 2-years cycle led to model overfitting due to a too large number of features. We thus investigated the effects of leaf nutrients, soil nutrients, soil pH, NPK dosage and weather indices in two separate models (1) seasonal weather indices for 2-year cycle, and (2) monthly weather indices during the reproduction year only.

We hope that it makes the manuscript clearer and will be happy to adjust our manuscript if any questions remain.